# *PxTret1-like* Affects the Temperature Adaptability of a Cosmopolitan Pest by Altering Trehalose Tissue Distribution

**DOI:** 10.3390/ijms23169019

**Published:** 2022-08-12

**Authors:** Huiling Zhou, Gaoke Lei, Yanting Chen, Minsheng You, Shijun You

**Affiliations:** 1State Key Laboratory for Ecological Pest Control of Fujian and Taiwan Crops, Institute of Applied Ecology, Fujian Agriculture and Forestry University, Fuzhou 350002, China; 2Joint International Research Laboratory of Ecological Pest Control, Ministry of Education, Fuzhou 350002, China; 3Ministerial and Provincial Joint Innovation Centre for Safety Production of Cross-Strait Crops, Fujian Agriculture and Forestry University, Fuzhou 350002, China; 4Institute of Plant Protection, Fujian Academy of Agricultural Sciences, Fuzhou 350013, China; 5BGI-Sanya, BGI-Shenzhen, Sanya 572025, China

**Keywords:** *Plutella xylostella*, CRISPR/Cas9, mutant strain, trehalose, temperature adaptability

## Abstract

Global warming poses new challenges for insects to adapt to higher temperatures. Trehalose is the main blood sugar in insects and plays an important role in energy metabolism and stress resistance. The transmembrane transport of trehalose mainly depends on the trehalose transporter (TRET1). *Plutella xylostella* (L.) is a worldwide agricultural pest; however, the effects of the trehalose transport mechanism and trehalose distribution in tissues on the development, reproduction and temperature adaptation of *P. xylostella* have yet to be reported. In this study, *PxTret1-like* was cloned and analyzed regarding its expression pattern. It was found that the expression of *PxTret1-like* was affected by ambient temperature. The knockout mutation of *PxTret1-like* was generated using a CRISPR/Cas9 system by targeted knockout. The trehalose content and trehalase activity of mutant *P. xylostella* increased at different developmental stages. The trehalose content increased in the fat body of the fourth-instar *P. xylostella*, and decreased in the hemolymph, and there was no significant change in glucose in the fat body and hemolymph. Mutant strains of *P. xylostella* showed a significantly reduced survival rate, fecundity and ability to withstand extreme temperatures. The results showed that *PxTret1-like* could affect the development, reproduction and temperature adaptability of *P. xylostella* by regulating the trehalose content in the fat body and hemolymph.

## 1. Introduction

Global climate change most probably lead to increased average temperatures and the frequent occurrence of extreme temperatures [1]. Temperature is a key factor that directly affects organisms [2]. Insects, as small ectotherms, are more susceptible to temperature [3]. Extreme temperatures may lead to altered phenotypes, such as body color, body weight and body size [4,5]. Extreme temperature changes also affect insects’ growth and development, which consequently change the population size distribution [6,7]. Insects develop a variety of mechanisms to protect themselves from these temperature stresses, and they respond to changes in temperature through phenotype, coordinated behavioral, physiological and phenological changes [8,9]. For instance, warming temperatures have reduced the wing and body size of *Dolichovespula sylvestrisin* in the past century [10], while *Danaus plexippus* avoid low temperatures in winter through long-distance flight [11]. Under high temperatures, *Bemisia argentifolii* rapidly accumulate sorbitol and mannitol, as these metabolites are associated with temperature tolerance [12].

Metabolites are considered as a bridge between the genome and the phenome [13], and they are important for adaptation to climate change [14]. Carbohydrates are viewed as energy-storage materials, and have been consistently reported as being crucial for temperature adaptation [12,15,16]. Trehalose is the primary blood sugar in insects, which accounts for 80–90% of sugar in the hemolymph [17]. Trehalose not only provides energy for insects’ growth, development, molting and metamorphosis, but it is also an important stress-indicating metabolite [18,19]. When the insect’s nutritional status or environment was changed, the trehalose concentration in the hemolymph can be freely regulated, stabilizing protein conformation and protecting insects from environmental influences [20]. In the codling moth in full diapause, trehalose is three times higher than that at the beginning of diapause, and the survivability at low temperatures was found to be stronger [21]. Moreover, the antifreeze activity is enhanced by trehalose in *Dendroides*
*canadensis* Latreille (Coleoptera Pyrochroidae) [22].

The trehalose metabolism pathway (synthesis, transport, decomposition) has been widely investigated in insects. Sucrose can be hydrolyzed into fructose and glucose in the gut after insect feeding [23]. Insects tend to intake excess sucrose, most of which is transformed into long-chain oligosaccharides and eliminated as honeydew [24,25]; the remaining part is used for energy metabolism and to maintain the osmotic balance [26,27]. Glucose is transported into the fat body via glucose transporters (GLUT), and is involved in the synthesis of trehalose by TPS/TPP [28,29]. Trehalose cannot directly cross the cell membrane [30], and it mainly depends on the specific trehalose transporter TRET1 to be transported between cells [31]. Kikawada et al. (2007) [32] isolated and characterized the functions of TRET1 from insects, and TRET1 was found to be involved in transporting trehalose synthesized in the fat body into the hemolymph. Trehalose is hydrolyzed to two glucose monomers by trehalase in the hemolymph, and transported in the blood to tissues to fulfil energy needs [31,33]. Currently, much of the research on the trehalose pathway focuses on trehalose synthesis and decomposition, while little is known about the mechanisms involved in trehalose transport [34].

The diamondback moth (DBM), *Plutella xylostella* (L.) (Lepidoptera: Plutellidae), is one of the most destructive pests for cruciferous vegetables [35], and the annual control costs of *P. xylostella* are estimated as approximately USD 4–5 billion worldwide [36]. *P. xylostella* can adapt quickly to new environments [37], and it is distributed across all continents except Antarctica [36]. Climate warming negatively affects insects and organisms at lower latitudes [38]; however, the high genetic tolerance of the diamondback moth will lead to the mitigation of the decline in its fitness, and it will continue to be rampant in the future [39,40]. Adaptation to extreme temperatures in *P. xylostella* has some guiding significance for predicting the dynamics and origins of populations, and formulating effective prevention and control measures [41]. Therefore, more attention should be paid to the temperature adaptation of *P. xylostella.*

In this study, we analyzed the molecular structure of PxTret1-like and the expression patterns of PxTret1-like at different temperatures. The knockout mutation of *PxTret1-like* was generated using a CRISPR/Cas9 system by targeted knockout. We analyzed the expression patterns of *TPS* and *TRE* of different ages and detected the different states of the hemolymph and fat body of *P. xylostella* and fourth-instar larvae regarding glucose and trehalose content, which indicated that *TRET1* participates in transporting trehalose synthesized in the fat body to the hemolymph. We established the age-stage-specific sex life table of different strains to clarify that the content of trehalose in different tissues of *P. xylostella* affects the insect’s development and reproduction. We performed a 42 °C stress treatment at different times in *P. xylostella* and detected the supercooling freezing and freezing point of pupae, which revealed that *P. xylostella* can rapidly adapt to temperature changes by regulating the tissue distribution of trehalose through *PxTret1-like*.

## 2. Results

### 2.1. Identification and Characterization of PxTret1-like

PxTret1-like had an open reading frame (ORF) with 1491 base pairs that encoded for 497 amino acids (aa). The PxTret1-like structure consisted of three exons and two introns, and the gene contained one highly conserved domain. *PxTret1-like* was found to be a hydrophobic protein with an isoelectric point of 9.45 and theoretical molecular weight of 54,884.81. The PxTret1-like secondary structure prediction suggested a helix and curl structure, and the transmembrane domain contained 12 transmembrane domains (Figure 1).

The phylogenetic tree showed that *PxTret1-like* and Pieris rapae were sister clades; Pieris brassicae, Colias croceus and Zerene cesonia in turn formed a monophyletic group. TRET1 was relatively conserved in Lepidoptera (Figure 2).

### 2.2. Expression Profiles of PxTret1-like in Different Stages and at Extreme Temperature

The expression profile indicated that the expression of *PxTret1-like* began in the egg and reached a maximum level at adult age (Figure 3a). The extreme temperature treatment of male and female adults for 30 min affected the expression profile of *PxTret1-like.* In the female, the expression level of *PxTret1-like* significantly increased at 32 °C, and significantly decreased at 35 °C, 38 °C and 40 °C. In the male, the expression level of *PxTret1-like* significantly decreased at 35 °C, 38 °C, 40 °C, 43 °C, 43 °C, 4 °C and −17 °C (Figure 3b,c).

### 2.3. Establishment of Homozygous PxTret1-like Knockout Strains

A mixture of Cas9 protein (200 ng/μL) and sgRNA (100 ng/μL) was injected into 100 newborn eggs of *P. xylostella*, and 62% (62/100) of eggs successfully developed into adults, which were regarded as the G0 generation. Sequencing the 62 G0 individuals indicated a mutation efficiency of 11.29% (7/62) in the target site at exon 3 of *PxTret1-like*. After the G1 generation of inbred and laid eggs, sequencing of the G1 generation showed three mutant types: *PxTret1-like*-1bp with 1 base deletion, *PxTret1-like*-2bp with 2 base deletions and *PxTret1-like*+14bp with 14 base insertions. The same mutation types were retained in G2 to develop a sibling hybrid strain. The sibling cross-pairs with a single homozygote were in G3 (aa×Aa) and the stable homozygous mutants were established at G4 (Figure 4).

### 2.4. Expression Profiles of TPS and TRE

Compared with the wild-type strain, the expression level of *TPS* in the three mutant strains significantly decreased in the different developmental stages (4th instar, pupae, female adult and male adult) (Figure 5a), and the expression levels of *TRE1-1* and *TRE1-2* significantly increased in the different developmental stages (Figure 5b,c).

### 2.5. Trehalose, Glucose and Trehalose Metabolic Enzymes

Compared with the wild-type strain, the trehalose content of the mutants significantly increased in the different developmental stages (fourth instar, pupae, female adult and male adult). The glucose content of the mutants significantly increased in the female adult and did not change significantly in fourth-instar, pupae and male adult insects. The trehalase activity of mutants significantly increased in the fourth-instar, female adult and male adult insects, but the trehalase activity of the mutant pupae did not change significantly (Figure 6a). In addition, we compared the content of trehalose and glucose in the hemolymph and fat body between wild-type insects and mutants. The content of trehalose significantly increased in the fat body but significantly decreased in the hemolymph. The glucose content of mutant and wild-type strains did not significantly change in the fat body and hemolymph (Figure 6b,c).

### 2.6. Age-Stage-Specific Sex Life Table

Values for the developmental time of each stage, preadult all, adult all, longevity all, adult pre-oviposition (APOP), total pre-oviposition (TPOP), female fecundity and female oviposition of the mutant and wild-type strains are listed in Table 1 and Table A1. The development times for each stage, preadult all, adult all, longevity all, TPOP and female fecundity were significantly decreased in the mutants compared with the wild-type strain. The net reproductive rate (*R*_0_), intrinsic rate of increase (*r*), finite rate of increase (*λ*) and generation time (*T*) of mutant and wild-type strains are listed in Table 2 and Table A1. The *R*_0_, *r* and *λ* of mutants were significantly shorter than those of the wild type, but the *T* of mutants was significantly longer. The survival rate of the mutant strain was lower than that of the wild-type strain (Figure 7i). The *l_x_* curve dropped only slightly during the early stages in the wild-type strain, which showed that the mortality rate at this stage was low. The age-stage-specific fecundity (*f_x_*) curve peak for mutants was lower than that of the wild-type strain. The curve of age-specific fecundity (*m_x_*) indicated that the female adults of the mutant strain began to reproduce earlier than the wild-type strain (Figure 7ii).

### 2.7. Response to Extreme Temperature

With the extension of the treatment time at 42 °C, the survival rate of male and female adults of the mutant strain was lower than that of the wild-type strain (Figure 8).

### 2.8. Supercooling Point and Freezing Point

We found that the supercooling points of mutant pupae (MU-1, MU-2, MU+14; −20.91 ± 0.69 °C, −20.26 ± 0.78 °C, −20.69 ± 0.65 °C) were significantly higher than those of the wild-type strain (−22.74 ± 0.33 °C). The freezing points of mutant pupae (MU-1, MU-2, MU+14; −7.65 ± 0.50 °C, −8.35 ± 0.52 °C, −5.55 ± 0.28 °C) were significantly higher than those of the wild-type strain (−10.21 ± 0.56 °C) (Figure 9).

## 3. Discussion

The amounts of insects or ectotherm are predicted to increase at mid- and high latitudes and to decrease at low latitudes by 2100, with global warming [38]. Insects’ ability to sense climate changes and respond to them is critical to their survival [42]. Trehalose transporter (TRET1) is a trehalose-specific facilitated transporter, which plays an important role in maintaining homeostasis and temperature stress tolerance in insects [34,43]. In this study, we used the CRISPR/Cas9 system to knock out *PxTret1-like*, indicating that TRET1 regulates the trehalose content in the fat body and hemolymph of *P. xylostella,* and we discovered that the trehalose content of the hemolymph and fat body affect the ability to respond to extreme temperatures in *P. xylostella*.

In this study, we found that *Tret1* in different insects has a similar amino acid sequence [44]. The secondary structures of PxTret1-like were predicted to be mainly α-helix and random coil, and PxTret1-like belonged to the Major Facilitator Superfamily (MFS). Similar to most sugar transporters [45], PxTret1-like has 12 predicted transmembrane domains.

In this study, we found *PxTret1-like* to be highly expressed in adult males and adult females. This might be because trehalose is the main energy source to support flight, and *PxTret1-like* expression is required to maintain flight in adults [46,47,48]. Moreover, trehalose has been broadly recognized in stress resistance. *Harmonia axyridis* (Pallas) (Coleoptera Coccinellidae) accumulated trehalose and showed inhibited trehalase activity at low temperatures [49]. *Drosophila melanogaster* Meigen (Diptera Drosophilidae) larvae accumulated trehalose in drought conditions, which worked in synergy with *TPS* and *TRE* [50]. The expression of *PxTret1-like* was significantly changed under high- or low-temperature stress treatments in female and male adults, which illustrated that the expression of *PxTret1-like* can respond to temperature changes.

To further investigate the role of PxTret1-like in the extreme temperature response, we created three *PxTret1-like* mutant lines via the CRISPR/Cas9 system. The content of trehalose was significantly increased in all the mutant lines (fourth-instar larvae, pupae, male adults and female adults), and was significantly increased in the fat body of fourth-instar larvae mutants, but the expression of *TPS* was significantly down-regulated. We speculated that the reduction in the trehalose transporter prevented trehalose synthesized in the fat body from being transported into the hemolymph, and the accumulation of trehalose in the fat body led to the down-regulation of *TPS* gene expression. Research found that the silencing of *TRET1a* in *Colaphellus bowringi* Baly (Coleoptera Chrysomelidae) elevated the trehalose content in the fat body and decreased the expression of some stress-related genes [48]. Trehalase is the only enzyme currently known to be capable of breaking down trehalose [51]. Trehalase can be classified as soluble or membrane-bound according to the transmembrane structure [52]. Soluble trehalase mainly breaks down intracellular trehalose, and membrane-bound trehalase mainly breaks down extracellular (mainly in food) trehalose [53]. Trehalose of *P. xylostella*, synthesized in the fat body and transported into the hemolymph, mainly breaks down by soluble trehalase. We found 2 soluble trehalase genes (*TRE1-1*, *TRE1-2*) in the *P. xylostella* genome. We found that the content of trehalose was significantly reduced in 4th-instar mutant *P. xylostella*, and the expression of trehalase genes was significantly increased in mutant (fourth instar, pupae, female adult and male adult) compared to wild-type *P. xylostella*. The same trend was observed for trehalase and trehalase genes in mutant *P. xylostella.* We concluded that the blocking of trehalose transport contributed to the lack of trehalose content in the hemolymph, and up-regulated the trehalase genes. It has been reported that the hemolymph trehalose of the *Treh^Cs1^* of mutant homozygotes is significantly increased in *Drosophila,* while the expression levels of *cTreh, sT**reh* and *Treh* are decreased [54]. Furthermore, there was no significant difference in the glucose content of mutant strains (4th instar, pupae and male adult) and wild-type strains, and there also was no significant difference in the glucose content of the hemolymph and fat body between the two strains. This suggests that, even in the absence of *PxTret1-like,* insects regulated the glucose in the fat body and hemolymph via other mechanisms—for example, other disaccharides (sucrose and lactose, etc.) were hydrolyzed to supplement this [55,56]. Glucose levels were elevated in mutant female adults; the reason for this may be that female mutants need energy to maintain normal life activities, such as mating and spawning [57]. Studies have shown that many insects need to use proteins and carbohydrates to synthesize sufficient lipids; as a result, insects can reach sexual maturity and lay eggs [58,59,60].

Previous studies have implicated important functions of insect development, metamorphosis and reproduction. For example, reduced trehalose content in the hemolymph of *Blattella germanica* (L.) (Blattodea Blattidae) led to prolonged spawning time [61]. A *Drosophila* mutant with trehalose-6-phosphate synthase (TPS1) was smaller than a wild type in body size, and the mutant strain could not grow normally with a low-sugar diet [62]. We constructed the age-stage-specific sex life table of the mutant and wild-type strains. The fecundity and survival of *P. xylostella* were declined in mutants, and the mutants’ population dynamic parameters (*R*_0_, *r*, *λ*) also declined, but the generation times (*T*) of mutants were longer than those of the wild type. We propose that the trehalose content of *P. xylostella* affected the developmental duration and female fecundity with *PxTret1-like* deletion. Research has found that slightly higher trehalose levels in fodder have a positive effect on *Harmonia axyridis* growth, development and reproduction. *Tret1a* and *Tret1b* silencing led to long-term diapause in *Colaphellus bowringi* [63]. We also found that the *PxTret1-like* mutants had reduced resilience to extreme temperatures. Under duress, accumulating large quantities of trehalose can eliminate high concentrations of reactive oxygen species in insects [64]. The mutants’ trehalose content in *P. xylostella* were increased in the whole body, but were reduced in the hemolymph. Silencing *AgTre**t1*, the trehalose content of *Anopheles gambiae* (Giles) (Diptera Culicidae) was reduced in the hemolymph, and *Anopheles gambiae* died sooner in hot-dry environments [65].

In this study, the CRISPR/Cas9 system was used to generate *PxTret1-like* knockout lines. We proved that the trehalose content of mutants was reduced in the hemolymph, which reduced the survival, fecundity and adaptability to of *P. xylostella* to extreme temperatures, according to the combined metabolite–transcript analysis in different strains, the age-stage-specific sex life table and extreme temperature stress.

This study clarified that knockout of *PxTret1-like* affected the development, reproduction and temperature adaptability of *P. xylostella* by regulating the trehalose content in tissues. However, trehalose was detected in mutants of *P. xylostella,* and there may be other trehalose transporter genes that regulate trehalose. Furthermore, glucose content did not change significantly in mutant and wild-type strains (4th instar, pupae and male adult), and further study is needed to discern whether other disaccharides’ hydrolysis replenishes the glucose. Trehalose is the main blood sugar in insects, which has been reported to play an important role in insects’ responses to abiotic stresses such as drought, pesticides and ultraviolet (UV) radiation. Whether the mutant of *PxTret1-like* reduces the adaptability to other abiotic stresses in *P. xylostella* remains to be clarified.

## 4. Materials and Methods

### 4.1. Insect Strains and Rearing

The *Plutella xylostella* wild-type (WT) strain was provided by the Institute of Zoology, Chinese Academy of Sciences. Larvae were reared on an artificial diet in 90 mm petri dishes, and adults were fed 10% honey solution in paper cups (10.4 cm × 7.3 cm × 8.5 cm) and maintained under 26 ± 0.5 °C, 60 ± 5% relative humidity, and a 12-h light/12-h dark photoperiod.

### 4.2. Gene Cloning

Total RNA from individuals was extracted using the Eastep^®^ Super Total 105 RNA Extraction Kit (Promega, Madison, WI, USA). Integrity of RNA samples was verified by gel electrophoresis with 2% agarose and staining with GelRed. Total RNA (2000 ng) was reverse-transcribed into complementary DNA (cDNA) using a Reverse-Transcription System Kit (Promega, Madision, WI, USA) in 20 μL total volume.

The sequence of *PxTret1-like* was retrieved from DBM genome data

(http://iae.fafu.edu.cn/DBM/index.php (accessed on 10 March 2021)). Primers for PCR cloning were designed with Oligo 7 (Table 3). PCR amplification was performed using the Phanta Max Super-Fidelity DNA Polymerase (Vazyme, Nanjing, China) with the following procedure: initial denaturation at 95 °C for 3 min, 34 cycles of denaturation at 95 °C for 30 s, annealing at 58 °C for 30 s, and then extension at 72 °C for 90 s, followed by the final extension at 72 °C for 5 min. PCR products were confirmed by 1% agarose gel electrophoresis and purified using a gel extraction kit (Omega Bio-Tek, Norcross, GA, USA), and then transformed into *E. coli* DH5a cells (TIANGEN, Beijing, China) for Sangon Biotech (Shanghai, China) sequencing.

### 4.3. Sequence Analysis and Phylogenetic Tree Construction

The cloned cDNA of *PxTret1-like* was compared with the DBM genome data to determine its exons and introns. The protein sequence and conserved domain were predicted by NCBI (https://blast.ncbi.nlm.nih.gov/Blast.cgi (accessed on 25 March 2021)). The molecular weight (*M*_W_) and isoelectric point (pl) of PxTret1-like were predicted using ExPASy (http://expasy.org/tools/dna.html (accessed on 25 March 2021)). The functional domain and transmembrane domain were predicted separately using PSIPRED (http://bioinf.cs.ucl.ac.uk/psipred (accessed on 25 March 2021)) and TMHMM (https://dtu.biolib.com/DeepTMHMM (accessed on 15 June 2022)).

The cDNA sequences of *PxTret1-like* were compared with the *Tret1* cDNA sequences of other insects in GenBank using the Blast tool on the NCBI website, and those of *P. xylostella* were constructed using maximum likelihood analysis with 1000 replicates in MEGA X.

### 4.4. Expression Profiling of PxTret1-like

#### 4.4.1. Expression Patterns of Different Stages

Total RNA was extracted from *P. xylostella* at different developmental stages (egg, one to four instars, pupae, female adult and male adult). The cDNA was obtained from an amount of 2000 ng total RNA using the Reverse-Transcription System Kit (Promega, Madision, WI, USA) and then diluted 1:10. The qRT-PCR primers are listed in Table 1, and the reference gene was RPL32^34^. The qPCR was performed with Bio-Rad IQ5 (Thermo, Waltham, MA, USA) using the GoTag^®^ qPCR Master Mix Kit (Promega, Madision, WI, USA). The samples were processed with the following procedure: initial denaturation at 95 °C for 10 min, 40 cycles of 15 s at 95 °C for denaturation, and 30 s at 60 °C for annealing, followed by a step for one cycle at 95 °C for 15 s, 60 °C for 60 s, and 95 °C for 15 s. RT-PCR was conducted with three replicates for each biological sample, for three biological replicates.

#### 4.4.2. Expression Patterns at Extreme Temperatures

The pupae were transferred individually to 1.5 mL tubes with wells, and then males and females at one day after elusion were treated at the following temperatures: (1) blank control (CK)—26 °C; (2) high temperature H1—32 °C for 30 min; (3) high temperature H2—35 °C for 30 min; (4) high temperatures H3—38 °C for 30 min; (5) high temperature H4—40 °C for 30 min; (6) high temperature H5—43 °C for 30 min; (7) low temperature L1—4 °C for 30 min; (8) low temperature L2—−17 °C for 30 min. Each treatment included five females and five males, and then samples were frozen immediately in liquid nitrogen measured as Section 4.4.1.

### 4.5. sgRNA Design and Synthesis

#### 4.5.1. sgRNA Design and Off-Target Analysis

The target site was designed in exon 3 of *PxTret1-like* based on the principle of CC(19 N)CC or GG(19 N)GG (Table 3), and the potential off-target effect of sgRNA was determined using the Cas-OFFinder design tool (http://www.rgenome.net/cas-offinder (accessed on 8 May 2021)).

#### 4.5.2. sgRNA Synthesis and Purification

The template of sgRNA for in vitro transcription was generated by PCR using the long unique oligonucleotides (Table 3) with the following reaction conditions: predenaturation at 95 °C for 3 min, 35 cycles of denaturation for 15 s at 95 °C, annealing for 30 s at 72 °C, and a final elongation at 72 °C for 5 min. PCR products were recovered using 2% TAE agarose gels and purified with a Gel Extraction Kit (Omega Bio-Tek, Norcross, GA, USA), and then transcribed in vitro using the HiScribe^TM^ T7 Quick High Yield RNA Synthesis Kit (New England Biolabs, Ipswich, MA, USA). The quality was checked with a spectrophotometer and 2% agarose gel electrophoresis, and the sgRNA was purified by the phenol/chloroform/isoamyl alcohol method [66].

#### 4.5.3. sgRNA/Cas9 Protein Microinjection

A mixture of 100 ng/μL Cas9 protein (GenScript, Nanjing, China) and 300 ng/μL gRNA was incubated at 37 °C for 20 min before microinjection, and then injected into newborn eggs (<20 min) using an Olympus SZX16 microinjection system (Olympus, Tokyo, Japan).

### 4.6. Mutation Screening

The microinjected eggs of *P. xylostella* that successfully developed into adults were defined as generation 0 (G0). The virgin G0 adults were sexed and individually mated with virgin wild-type adults to generate first transgenic lines (G1), and then the gDNA was extracted via the TiANamp Genomic DNA Kit (TIANGEN, Beijing, China). After this, all of the G0 individuals were directly sequenced with specific primers and reserved mutations (Table 1). The reserved G1 moths were self-fertilized in single pairs to generate G2 progeny, and then the lines with the same allelic mutation were reserved. The reserved G2 moths were self-fertilized in single pairs to generate G3 progeny until the homozygous mutations were reserved, as in the above method.

### 4.7. Expression Patterns of TPS and TRE

The total RNA was extracted from wild-type and mutant different developmental stages (four instar, pupae, female adult and male adult), and cDNA was synthesized using the Reverse-Transcription System Kit (Promega, Madison, WI, USA) to detect the expression of *TPS* (trehalose-6-phosphate synthase) and *TRE* (trehalase). The qRT-PCR primers of *TPS*, *TRE-1* and *TRE-2* are shown in Table 1. The detection method was the same as described in Section 4.4.

### 4.8. Determination of the of Trehalose, Glucose and Trehalose Metabolic Enzymes

The 20 mg insects (4th instar, pupae, female adult and male adult) in Section 4.7 were measured for the content of trehalose, glucose and trehalose metabolic enzymes. The wild-type and mutant fourth-instar larvae were examined for the content of trehalose and glucose in the hemolymph and adiposomes. Trehalose and glucose quantification: The hemolymph (5 μL) and adiposomes (50 heads), at different developmental stages (20 mg), were homogenized in a 1.5 mL polyethylene centrifugation tube containing 250 μL 0.25 M Na_2_CO_3_. The samples were incubated at 95 °C for 20 min to inactivate all enzymes. The pH of the samples was adjusted to 5.2 by adding 600 μL 0.2 M sodium acetate and 150 μL 1 M acetic acid, and then the samples were centrifuged at 25 °C at 13,800× *g* for 10 min. Next, 100 μL of the suspension was incubated overnight with 2 μL porcine kidney trehalase (Sigma-Aldrich, St Louis, MO, USA) at 37 °C. The amount of glucose in 20 μL of the treated supernatant was determined by the Glucose Assay Kit (Comin, Suzhou, China). The glucose concentration in the sample was corrected by deducting the glucose concentration in the supernatant before trehalase treatment [67,68]. Trehalose metabolic enzyme quantification: The enzyme activity of trehalose was measured using the 3,5-dinitrosalicylic acid assay according to the trehalase kit (Comin, Suzhou, China). Experiments were repeated five to six times.

### 4.9. Age-Stage-Specific Sex Life Table

The wild-type and mutant strain were used to construct an age-stage-specific sex life table under the same conditions as in Section 2.1. In total, 120 eggs (<30 min) were randomly chosen and placed in 30 mm petri dishes and fed an artificial diet. The artificial diet was changed every two days. The neonates were counted, and the survival and development times were recorded daily. The pupae were transferred individually to 1.5 mL tubes with wells and a pair of newly emerged male and female *P. xylostella* was placed in a plastic cup (29 mm × 38 mm × 32 mm) with a well. The papers with grooves were placed in cups as oviposition substrates. Each female that laid eggs was analyzed until it died. Parameters such as fecundity, survivorship and oviposition period were recorded. Each newly hatched larva was regarded as a replicate.

Life history data of wild-type and mutants were analyzed using the TWOSEX-MSChar computer program (http://140.120.197.173/ecology/prod2.htm (accessed on 15 February 2022)) [69]. It calculated the age-stage-specific survival rate (*S_xj_*), the age-stage-specific fecundity (*f_xj_*), the age-specific-survival rate (*l_x_*), the age-specific fecundity (*m_x_*), the net reproductive rate (*R*_0_), the intrinsic rate of increase (*r*), the finite rate (*λ*) and the mean generation time (*T*). The formulas for the parameters are given in the following equations:$$l_{x}=\sum_{j=1}^{m}Sxj$$
$$m_{x}=\frac{\sum_{j=l}^{m}S_{xj}f_{xj}}{\sum_{j=l}^{m}S_{xj}}$$
$$R_{0}=\sum_{x=0}^{\infty }l_{x}m_{x}$$
$$\sum_{x=0}^{\infty }e^{-r(x+1)}l_{x}m_{x}=1$$
$$\lambda =e^{r}$$
$$T=(\ln R_{0})/r$$

The standard errors of the life history parameters and variances were calculated 100,000 times using a bootstrap procedure.

### 4.10. Response to Extreme Temperature

The newly emerged female and male *P. xylostella* of wild-type and mutant strains were transferred individually to 1.5 mL tubes with wells. Here, 20 females or males were regarded as a replicate, and each treatment was repeated four times. The temperature treatments were applied as follows: the samples were treated under 42 °C for different times (30 min, 60 min, 90 min, 120 min, 150 min). The female and male moths were transferred to 26 °C for 24 h after treatment and we counted the surviving moths. The moths’ limb (antennae, feet and beak) activity was considered to indicate that they were alive.

### 4.11. Supercooling Point and Freezing Point

Two-day-old pupae of wild-type and mutant moths were randomly selected to detect the supercooling point and freezing point. The thermistor temperature sensing probe (Omega Bio-Tek, Norcross, GA, USA) was placed fully in contact with the pupa using conductive tape. The pupa was then placed in a 1.5 mL tube mounted with cotton. The tube was placed in a −65 °C ultra-low-temperature refrigerator, and the refrigerator cooling rate was 0.5 °C/s. Temperature change was observed every second, and the temperature reversal point was recorded [70].

### 4.12. Statistical Analyses

The expression of *PxTret1-like* in different stages was calculated based on the 2^−ΔCt^ method. The expression of other factors was calculated by the 2^−ΔΔCt^ method. The results were analyzed with the use of SPSS (19.0). A Shapiro–Wilk test was performed to detect the normally distributed data. If the data were normally distributed, an independent-sample *t*-test was used to test the significant differences in *PxTret1-like* gene expression at the different stages and under different temperature treatments. If the data were non-normally distributed, the Mann–Whitney U-test was used to test the *PxTret1-like* gene expression.

## Figures and Tables

**Figure 1 ijms-23-09019-f001:**
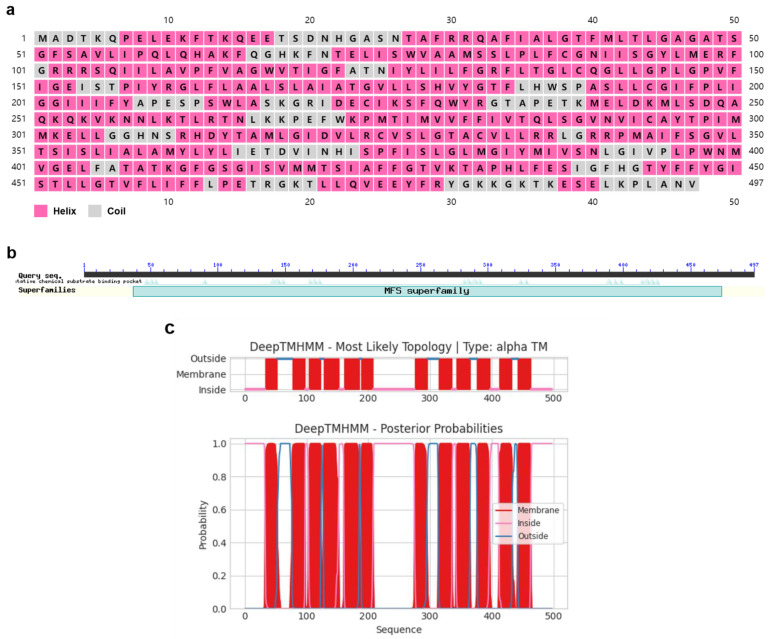
Characterization of *Plutella xylostella PxTret1-like* gene. (**a**) Secondary structure of PxTret1-like. Pink represents helix; grey represents coil. (**b**) Domain of PxTret1-like. (**c**) Transmembrane domain of PxTret1-like. Red line represents membrane; pink line represents inside; blue line represents outside.

**Figure 2 ijms-23-09019-f002:**
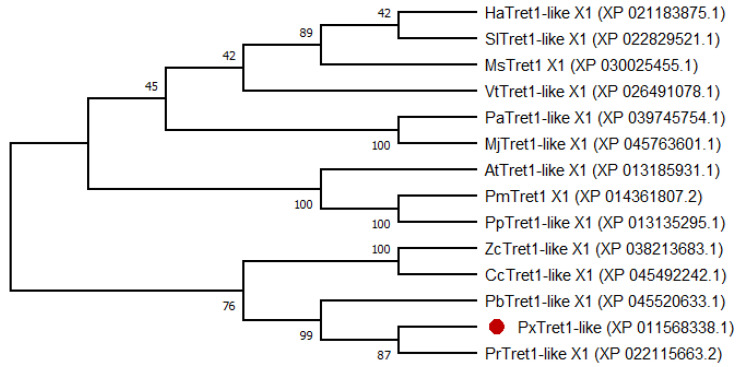
Phylogenetic tree of *PxTret1-like* from different insects based on amino acid sequences. *PxTret1-like* is marked by a red dot. *Helicoverpa armigera*: HaTret1-like X1; *Spodoptera litura*: SlTret1-like X1; *Manduca sexta*: MsTret1 X1; *Vanessa tameamea*: VtTret1-like X1; *Pararge aegeria*: PaTret1-like X1; *Maniola jurtina*: MjTret1-like X1; *Amyelois transitella*: AtTret1-like X1; *Papilio machaon*: PmTret1 X1; *Papilio polytes*: PpTret1-like X1; *Zerene cesonia*: ZcTret1-like X1; *Colias croceus*: CcTret1-like X1; *Pieris brassicae*: PbTret1-like X1; *Plutella xylostella*: *PxTret1-like*; *Pieris rapae*: PrTret1-like X1.

**Figure 3 ijms-23-09019-f003:**
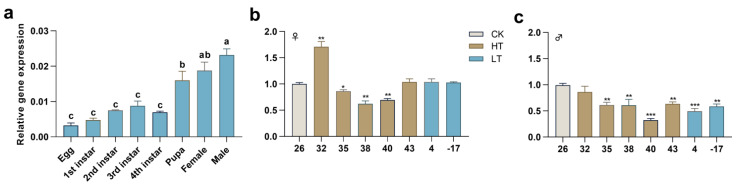
The expression patterns of *PxTret1-like* in *Plutella xylostella*. (**a**) Expression development patterns of *PxTret1-like*; (**b**) expression of *PxTret1-like* in female adults treated at different temperatures (32 °C, 35 °C, 38 °C, 40 °C, 43 °C, 4 °C, −17 °C); (**c**) expression of *PxTret1-like* in male adults treated at different temperatures (32 °C, 35 °C, 38 °C, 40 °C, 43 °C, 4 °C, −17 °C). The Student–Newman–Keuls test was used to analyze the expression of *Pxtret1-like* at different developmental stages. The independent-samples *t*-test was used to analyze the expression of *Px**Tret1-like* in female and male adults subjected to stress at different temperatures. Error bars represent standard error of the mean. Different letters indicate significant differences in expression levels. * indicates *p* < 0.05; ** indicates *p* < 0.01; *** indicates *p* < 0.001.

**Figure 4 ijms-23-09019-f004:**
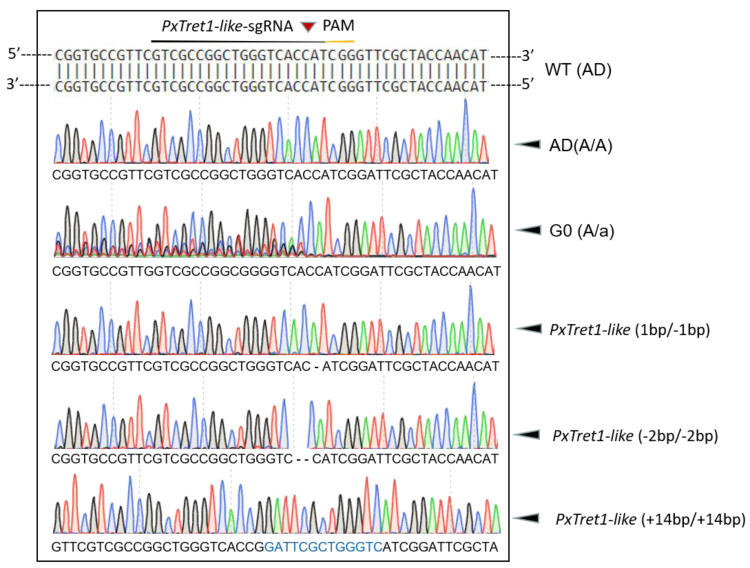
Targeted mutation of *Plutella xylostella PxTret1-like* mediated by CRISPR/Cas9. The sgRNA target sequence of *PxTret1-like* was identified in exon 3 and is underlined in black; the PAM sequence adjacent to the sgRNA target sequence is also underlined in yellow; and the cleavage site is indicated with a red inverted triangle. Green peak represents A; red peak represents T; blue peak represents C; and black peak represents G. A stretch of typical multiple peaks of PCR products directing sequencing was the main characteristic of mutated G0 individuals. Deleted bases are shown as dashes, and the inserted bases are indicated in blue.

**Figure 5 ijms-23-09019-f005:**
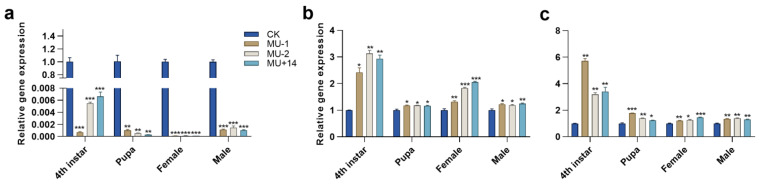
Effects of *Pxtret1-like* knockout on the expression of *TPS* and *TRE* in *Plutella xylostella* at different developmental stages. (**a**) The expression of *TPS*; (**b**) the expression of *TRE1-1*; (**c**) the expression of *TRE1-2.* CK: wild-type strain; MU-1: *PxTret1-like*-1 bp mutant; MU-2: *PxTret1-like*-2 bp mutant; MU+14: *PxTret1-like+*14 bp mutant. Error bars represent standard error of the mean. An ndependent-samples *t*-test was used. * indicates *p* < 0.05; ** indicates *p* < 0.01; *** indicates *p* < 0.001.

**Figure 6 ijms-23-09019-f006:**
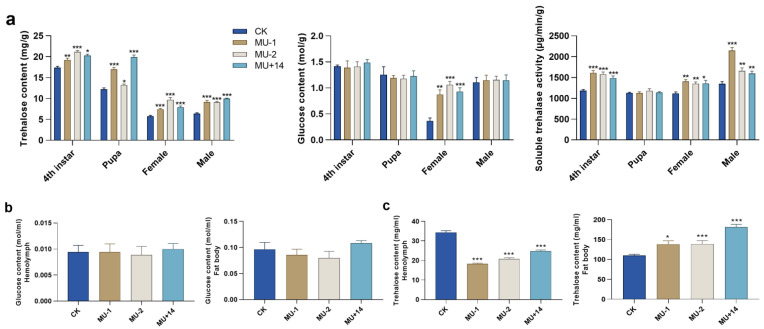
Effects of *Px**Tret1-like* knockout on trehalose content, glucose content and trehalase activity. (**a**) Trehalose content, glucose content and trehalase activity in 4th-instar, pupae, female adults and male adults of *Plutella xylostella*; (**b**) trehalose content in hemolymph and fat body; (**c**) glucose content in hemolymph and fat body. Independent-samples *t*-test was used. Error bars represent standard error of the mean. Independent-samples *t*-test was used. * indicates *p* < 0.05; ** indicates *p* < 0.01; *** indicates *p* < 0.001.

**Figure 7 ijms-23-09019-f007:**
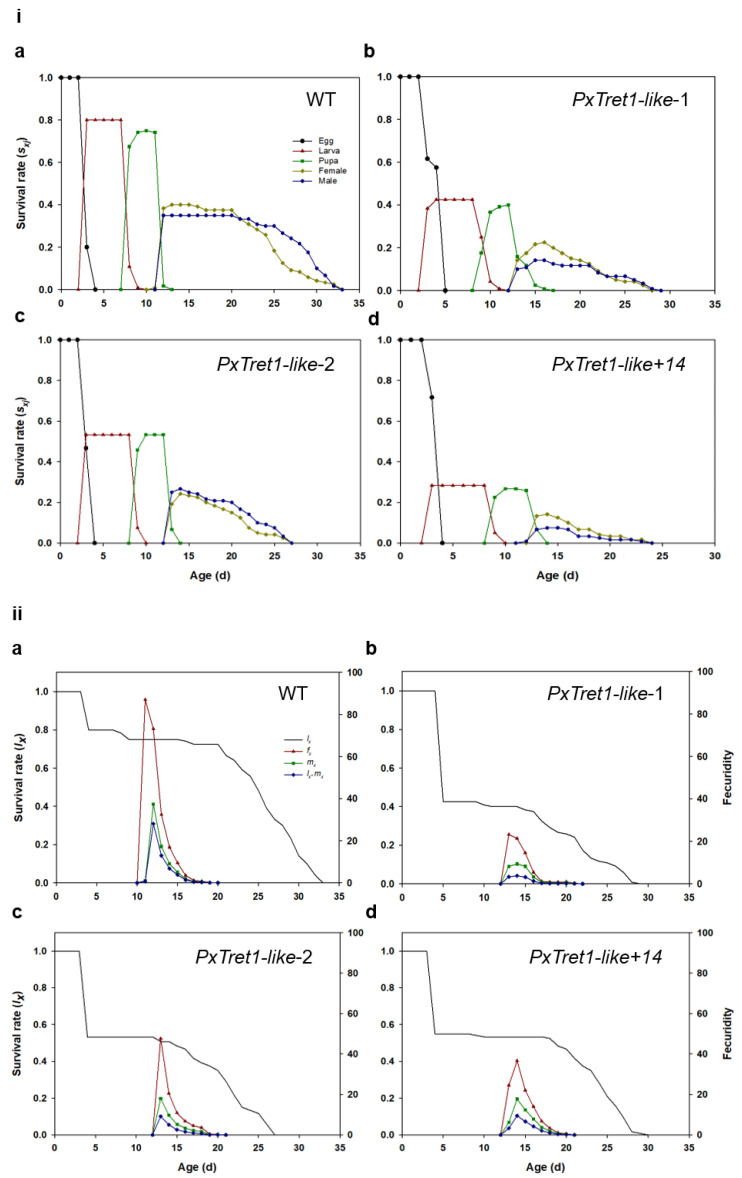
Effects of *Px**T**ret1-like* knockout on age-stage characteristics of *Plutella xylostella*. (**i**) Age-stage survival rates (*S_xj_*) of *Plutella xylostella* in different strains and in different generations; (**ii**) age-specific survival rates (*l_x_*), female age-stage specific fecundity (*f_x_j*) and age-specific fecundity of total population (*m_x_*) of *Plutella xylostella* in different strains and in different generations. (**a**): wild-type strain; (**b**–**d**): mutant of *PxTret1-like*.

**Figure 8 ijms-23-09019-f008:**
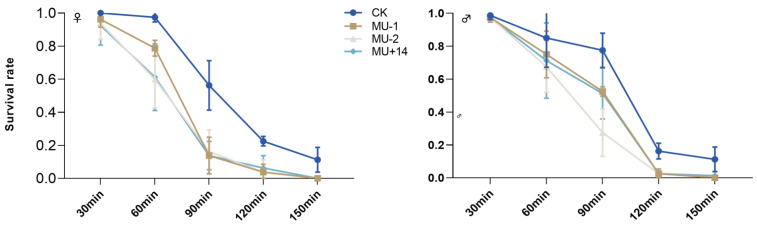
Effects of *Px**T**ret1-like* knockout on the survival of adults exposed to extreme temperatures for different times. Error bars represent standard error of the mean. Data represent 20 female adults or male adults per repetition and each replication was repeated four times.

**Figure 9 ijms-23-09019-f009:**
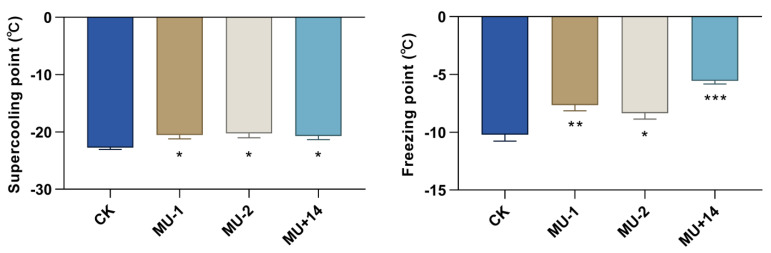
Effects of *Px**Tret1-like* knockout on supercooling point and freezing point. Independent-samples *t*-test was used, and at least 30 pupae of each strain were detected. Error bars represent standard error of the mean. * indicates *p* < 0.05; ** indicates *p* < 0.01; *** indicates *p* < 0.001.

**Table 1 ijms-23-09019-t001:** Developmental time, longevity, APOP, TPOP, fecundity, and Oviposition on different *PxTret1-like* mutants and wild-type strain in different generations.

Parameter	WT	*PxTret1-like*-1	*PxTret1-like*-2	*PxTret-like*+14
Egg (d)	3.00 ± 0.00	3.10 ± 0.04 *	3.00 ± 0.00	3.00 ± 0.00
Larva (d)	5.11 ± 0.04	6.58 ± 0.08 *	6.14 ± 0.04 *	6.44 ± 0.08 *
Pupa (d)	3.90 ± 0.04	4.08 ± 0.10 *	3.98 ± 0.02	4.16 ± 0.05 *
Preadult All (d)	12.01 ± 0.02	13.74 ± 0.16 *	13.13 ± 0.04 *	13.59 ± 0.08 *
Adult F (d)	13.27 ± 0.57	8.00 ± 0.73 *	8.07 ± 0.62 *	10.50 ± 0.56 *
Adult M (d)	16.60 ± 0.49	9.21 ± 1.06 *	9.19 ± 0.65 *	11.07 ± 0.53 *
Adult All (d)	14.82 ± 0.42	8.49 ± 0.61 *	8.66 ± 0.45 *	10.75 ± 0.39 *
longevity All (d)	21.36 ± 0.93	11.97 ± 0.80 *	13.27 ± 0.84 *	14.94 ± 0.95 *
APOP (d)	0.00 ± 0.00	0.53 ± 0.19 *	0.19 ± 0.09	0.17 ± 0.08 *
TPOP (d)	11.98 ± 0.02	13.83 ± 0.23 *	13.33 ± 0.13 *	13.83 ± 0.13 *
Oviposition (d)	4.93 ± 0.24	3.26 ± 0.41 *	4.07 ± 0.38	4.60 ± 0.28
Fecundity (eggs)	151.9 ± 7.70	75.95 ± 13.88 *	88.04 ± 9.93 *	92.29 ± 7.45 *

Note: The differences between treatments were evaluated by using a paired bootstrap test. d represents day. Values are means ± standard errors; Asterisk (*) indicates *p* < 0.05 between mutant and wild-type strain.

**Table 2 ijms-23-09019-t002:** Population parameters of different *PxTret1-like* mutants and wild-type strain in different generations.

Parameter	WT	*PxTret1-like*-1	*PxTret1-like*-2	*PxTret-like*+14
*r* (d^−1^)	0.29 ± 0.01	0.16 ± 0.02 *	0.20 ± 0.01 *	0.21 ± 0.01 *
*R*_0_ (eggs/female)	54.43 ± 7.19	12.03 ± 3.31 *	19.81 ± 4.03 *	26.92 ± 4.38 *
*T* (d)	13.68 ± 0.05	15.84 ± 0.23 *	14.94 ± 0.01 *	15.70 ± 0.11 *
*λ* (d^−1^)	1.34 ± 0.01	1.18 ± 0.02 *	1.22 ± 0.02 *	1.23 ± 0.01 *

Note: The intrinsic rate of increase (*r*), net reproductive rate (*R*_0_), finite rate of increase (*λ*), and generation time (*T*). The data of treatments were compared by using a paired bootstrap test. d represents day. Values are means ± standard errors; Asterisk (*) indicates *p* < 0.05 between mutant and wild-type strain.

**Table 3 ijms-23-09019-t003:** Primers used in this study.

	Primer Name	Primer Sequence 5′-3′	Position
PCR	CDS F	ATGGCGGACACGAAACAGC	1–19
	CDS R	TCAAACATTAGCTAAAGGCTTCAATTC	1468–1494
Quantitative PCR	Q-PxTret1-like F	GCTTCTCCGCCGTGCTTATCC	2016–2036
	Q-PxTret1-like R	AGAACAGTGGAAGCGATGACA	2307–2327
	Q-PxTPS F	GTCCGACCCCAATGACAACACG	827–848
	Q-PxTPS R	CAGCGGCCAGAAGGTCCC	2913–2930
	Q-PxTRE1-1 F	CAGCAGCAGCTCTACTCCATCATC	6369–6392
	Q-PxTRE1-1 R	GTGTCGCGCATCTCCGACAG	10,015–10,034
	Q-PxTRE1-2 F	CTCCGAGGACTACGAGAATGCG	885–906
	Q-PxTRE1-2 R	GATGGCGTTCTGGTCGACGG	1087–1106
sgRNA synthesis	SgRNA F ^a^	TAATACGACTCACTATAGGGTCGCCGGCTGGGTCACCATG	2405–2424
		TTTTAGAGCTAGAAATAGCAAGTTAAAATAAGGCTAGTCC	
	sgRNA-ComR ^b^	AAAAGCACCGACTCGGTGCCACTTTTTCAAGTTGATAAC	-
		GGACTAGCCTTATTTTAACTTGCTATTTCTAGCTCTAAAA	

F represents Forward primer; R represents Reverse primer. ^a^ T7 promoter sequence is single underline. Target site is double underline. ^b^ It is a universal reverse primer.

## Data Availability

The data presented in this study are available on request from the corresponding author.

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
