# Peer review of "PxTret1-like* Affects the Temperature Adaptability of a Cosmopolitan Pest by Altering Trehalose Tissue Distribution"

_ijms, 2022, doi:10.3390/ijms23169019_

Round 1

Reviewer 1 Report

Zhou et al submitted a manuscript titled "PxTret1-like affects the temperature adaptability of a cosmopolitan pest by altering trehalose tissue distribution" for publication in IJMS. The work is interesting, however the writing requires some improvement and few corrections.

Line 14: Global temperature is well-known concept, so change the first sentence to "Global warming poses new challenges on insects to adapt to higher temperatures."

Figure 1 is about secondary structure prediction and transmembrane domains. The figure description requires more explanation of all elements of the figure.

Remove the legend that gives color code of Figure 1a. In figure description, just mention in words, what pink and grey highlights represent.

Line 230: "* indicates P < 0.05; **indicates P < 0.01; ***indicates P < 0.001." This part can be deleted as it is universal standard.

After the authors address these issues, the manuscript can be accepted for publication.

Author Response

Dear reviewer,

Many thanks for all your kind advice and comments concerning our manuscript (ijms-1828200) entitled “PxTret1-like affects the temperature adaptability of a cosmopolitan pest by altering trehalose tissue distribution”. The comments are all valuable and very helpful for revising and improving our paper, as well as important in clarifying the significance of our research. We have studied the comments carefully, and made revision in response to the comments, which are highlighted using the track changes mode in the text of our manuscript. A point-by-point response to the comments is presented as follows:

Comments and Suggestions for Authors

Zhou et al submitted a manuscript titled "PxTret1-like affects the temperature adaptability of a cosmopolitan pest by altering trehalose tissue distribution" for publication in IJMS. The work is interesting, however the writing requires some improvement and few corrections.

  1. Line 14: Global temperature is well-known concept, so change the first sentence to "Global warming poses new challenges on insects to adapt to higher temperatures."

Response: As suggested, we have changed the first sentence (line 15) to “Global warming poses new challenges on insects to adapt to higher temperatures.” 

  1. Figure 1 is about secondary structure prediction and transmembrane domains. The figure description requires more explanation of all elements of the figure. Remove the legend that gives color code of Figure 1a. In figure description, just mention in words, what pink and grey highlights represent.

Response: As suggested, the legend of Figure 1 has been revised as set out below: “Characterization of Plutella xylostella PxTret1-like gene. (a) Secondary structure of PxTret1-like. Pink represents helix; grey represents coil. (b) Domain of PxTret1-like. (c) Transmembrane domain of PxTret1-like. Red line represents membrane; pink line represents inside; blue line represents outside.” In addition, Figure 1a has been changed as follows.

  1. Line 230: "* indicates P < 0.05; **indicates P < 0.01; ***indicates P < 0.001." This part can be deleted as it is universal standard.

Response: As suggested, we have deleted "* indicates P < 0.05; **indicates P < 0.01; ***indicates P < 0.001 (line 241)."

Reviewer 2 Report

This is an interesting paper. Many species have variability that allows them to get over difficult periods from the climatic point of view.

Thus, global warming may be involved in this kind of paper, but it should be not the main objective of the research. Thus, do not emphasize excessively the relationship between genetic variability and global warming.

You will find here the manuscript with some minor revisions.

Author Response

Dear reviewer,

Many thanks for all your kind advice and comments concerning our manuscript (ijms-1828200) entitled “PxTret1-like affects the temperature adaptability of a cosmopolitan pest by altering trehalose tissue distribution”. The comments are all valuable and very helpful for revising and improving our paper, as well as important in clarifying the significance of our research. We have studied the comments carefully, and made revision in response to the comments, which are highlighted using the track changes mode in the text of our manuscript. A point-by-point response to the comments is presented as follows:

Comments and Suggestions for Authors

This is an interesting paper. Many species have variability that allows them to get over difficult periods from the climatic point of view.

Thus, global warming may be involved in this kind of paper, but it should be not the main objective of the research. Thus, do not emphasize excessively the relationship between genetic variability and global warming.

Response: As suggested, we have revised the expression about the relationship between genetic variability and global warming (line 78-79) as follows: “P. xylostella can adapt quickly to new environments [37], and it is distributed across all continents except Antarctica [36].”

Details:

  1. Line33: change “will continue to lead” to “most probably lead”

Response: We have changed “will continue to lead” to “most probably lead” (line 35).

  1. Line41: change “behavioral and physiological changes” to “behavioral, physiological and phenological changes”

Response: We have changed “behavioral and physiological changes” to “behavioral, physiological and phenological changes” (line 43).

  1. Line 57: change “Dendroides Canadensis” to “Dendroides canadensis Latreille (Coleoptera Pyrochroidae)”.

Response: We have changed “Dendroides Canadensis” to “Dendroides canadensis Latreille (Coleoptera Pyrochroidae)” (line 60).

  1. Line 79: please, give the English name when you cite for the first time the species (Diamondback moth)

Response: As suggested, we have given the English name (Diamondback moth) when we cite for the first time the species (line 76).

  1. Line 249: change “Harmonia axyridis” to “Harmonia axyridis (Pallas) (Coleoptera Coccinellidae)”

Response: We have changed “Harmonia axyridis” to “Harmonia axyridis (Pallas) (Coleoptera Coccinellidae)” (line 260-261).

  1. Line 250: change “Drosophila melanogaster” to “Drosophila melanogaster Meigen (Diptera Drosophilidae)”

Response: We have changed “Drosophila melanogaster” to “Drosophila melanogaster Meigen (Diptera Drosophilidae)” (line 262).

  1. Line 263: change “Colaphellus bowringi” to “Colaphellus bowringi Baly (Coleoptera Chrysomelidae)”

Response: We have changed “Colaphellus bowringi” to “Colaphellus bowringi Baly (Coleoptera Chrysomelidae)” (line 275-276).

  1. Line 292: change “Blattella germanica” to “Blattella germanica (L.) (Blattodea Blattidae)”

Response: We have changed “Blattella germanica” to “Blattella germanica (L.) (Blattodea Blattidae)” (line 305).

  1. Line 306: change “Anopheles gambiae” to “Anopheles gambiae (Giles) (Diptera Culicidae)”.

Response: We have changed “Anopheles gambiae” to “Anopheles gambiae (Giles) (Diptera Culicidae)” (line 320).